# Synthesis and Investigation of the Properties of a Branched Phthalonitrile Containing Cyclotriphosphazene

**DOI:** 10.3390/molecules29235668

**Published:** 2024-11-29

**Authors:** Dengxun Ren, Zexu Fan, Jiaqu Zhang, Yi Xu, Xianzhong Tang, Mingzhen Xu

**Affiliations:** School of Materials and Energy, University of Electronic Science and Technology of China, Chengdu 610054, China; rendenxun2008@126.com (D.R.); fanzexu150@163.com (Z.F.); 19908094739@163.com (J.Z.); yogurtxy@163.com (Y.X.); txzhong@uestc.edu.cn (X.T.)

**Keywords:** cyclotriphosphazene, branched phthalonitrile, curing reaction, thermomechanical properties, thermal decomposition analysis

## Abstract

To study the properties of cyclotriphosphazene (CTP)-containing phthalonitriles, a branched phthalonitrile containing CTP (CTP–PN) with self-catalytic behavior was designed and synthesized. The structure of CTP–PN was characterized by FT–IR (Fourier transform infrared spectroscopy), MS (mass spectroscopy), ^1^H–NMR (proton nuclear magnetic resonance spectroscopy), and ^13^C–NMR (carbon nuclear magnetic resonance spectroscopy). Then, the curing reaction of CTP–PN was studied using DSC (differential scanning calorimetry) and DRA (dynamic rheological analysis). The results show that the curing reaction of CTP–PN is initiated at 200 °C. Additionally, the change in the viscosity of CTP–PN as a function of the temperature was investigated. After curing at different temperatures, the generated structures were characterized by FT–IR. The fracture morphology and thermomechanical properties of cured CTP–PN were scanned and studied using SEM (scanning electron microscopy) and TMA (thermomechanical analysis), respectively. The results demonstrate that CTP–PN exhibits a smooth fracture surface and possesses a relatively low CTE (coefficient of linear thermal expansion) of approximately 25 ppm/°C at 285 °C. A *T*_d5%_ (temperature at which 5% weight loss occurs) of as high as 405 °C can be obtained for cured CTP–PN, and its char yield at 800 °C exceeds 70% in N_2_. FT–IR and XPS (X-ray photoelectron spectroscopy) were used to study the thermal decomposition of cured CTP–PN, indicating that it remains stable below 350 °C. With an increasing temperature, there is decomposition first of CTP and P–NH–Ph and C–O–C bonds (>350 °C) and then nitrogen-containing aromatic heterocycles (>500 °C), ultimately resulting in the formation of P-containing residual char.

## 1. Introduction

High-performance polymer-based materials are advanced materials essential for the development of high-tech fields [1]. Of the various types of polymers, thermosetting resins exhibit relatively low fabrication temperatures, good dimensional stability, a high glass transition temperature, and good corrosion resistance due to their natural formation of three-dimensional crosslinking networks. Phthalonitrile, due to the generation of isoindoline, triazine rings, and phthalocyanine rings from the ring-forming reaction of nitrile groups, exhibits favorable mechanical properties, outstanding thermal stability, a high glass transition temperature, and flame retardancy [2,3]. Thus, phthalonitrile and its composites can be used as encapsulation materials for electronics in harsh environments [4], high-performance electronic materials (low or high *D*_k_) [3,5], flame-retardant materials (e.g., marine and aerospace applications), and high-performance structural materials [2,6].

Phthalonitrile monomers synthesized from diphenols and 4-nitrophthalonitrile exhibit relatively high melting points (>200 °C) and very slow curing reaction rates; after the treatment of monomers at 260–290 °C for several days, an increase in viscosity is observed [7,8,9]. However, these drawbacks of phthalonitrile-based materials limit their fabrication and applications. Thus, methods involving the addition of curing agents are often used to accelerate phthalonitrile polymerization. Curing agents such as metal salts (ZnCl_2_, AlCl_3_), organic amines, phenols, organic acids, and inorganic/organic mixed curing agents (CuCl/4,4′-diaminodiphenylsulfone (DDS) and ZnCl_2_/DDS; melamine (ME)/ZnCl_2_) have been widely investigated and used [8,10,11,12]. Additionally, other novel curing agents like multiple-SO_3_H functional ionic liquid [13] and methyl tetrahydrophthalic anhydride end-capped imide compounds [10] have been developed and studied. However, the currently available curing agents have shortcomings of a poor compatibility with phthalonitrile, poor matrix dispersion, pyrogenic decomposition, and negative effects on the thermal-oxidative stability of cured phthalonitrile [11,14]. To overcome the drawbacks of adding curing agents, researchers have developed various phthalonitriles with the property of self-catalysis. Most of these self-promoting phthalonitriles have been designed based on two principles: (1) the incorporation of active hydrogen-containing structures into molecules and (2) the generation of structures from introduced functional groups that can catalyze the curing reaction of nitrile groups [15]. Hydrogen-containing structures such as novolac-phthalonitrile resin with phenolic hydroxyl groups [16,17,18], a secondary amine group, and imidazole-group-containing phthalonitrile have been designed and prepared [14,19]. In addition to the active hydrogen, which has the ability to promote the curing reaction of nitrile groups, other catalytic mechanisms have also been proposed. In particular, Yang’s group has conducted numerous works aimed at promoting the curing reaction of nitrile groups. Based on their research on the curing reactions of phthalonitrile and/or phenylethynyl end-capped imide compounds [20], blends of phthalonitrile and alicyclic imide compounds [21], blends of phthalonitrile and methyl tetrahydrophthalic anhydride end-capped imide compounds [10], and alicyclic di-imide phthalonitrile model monomers [22], they found that the curing reaction of phthalonitrile could be accelerated by blending it with alicyclic imide moieties [21]. Additionally, a free radical mechanism for phthalonitrile curing was confirmed [23,24]. Thus, it can be seen that many novel phthalonitrile monomers with self-catalysis properties have been designed that solve the problems caused by adding catalysts.

Unlike blending modification methods, molecular structure modification is a method for balancing the intrinsic relationships between structures and properties at the molecular level. By introducing different structures into phthalonitrile molecules, phthalonitrile resins with a variety of properties can be produced. For example, methods for introducing flexible aromatic ether segments (containing ether bonds) [25], aliphatic chain segments [19], silicon-containing segments [26], and asymmetric structures [24] into phthalonitrile molecular chains have been developed for lowering the melting point of phthalonitrile. Moreover, branched phthalonitrile resins have also been designed and studied, although there are relatively few reports. Zu et al. prepared several kinds of branched phthalonitrile resins, such as branched poly(biphenyl ether triphenyl-*s*-triazine) phthalonitriles (*b*-PBP-Ph) and 1,3,5-tris(4-fluorobenzoyl)benzene (TB)-based branched phthalonitrile resins, in which the high density of nitrile groups led to the higher crosslinking efficiency of the precursors [27,28]. Kong et al. prepared a branched phthalonitrile (BPN) using boric acid, resorcinol, and 4-nitrophthalonitrile. Due to the high density of nitrile groups, the cured BPN exhibits excellent thermostability [29]. Thus, designing branched phthalonitrile resins is a strategy for endowing phthalonitrile with outstanding thermal stability by increasing the density and crosslinking efficiency of the nitrile groups.

Cyclotriphosphazene (CTP) is a ring compound consisting of three phosphorus and three nitrogen atoms in an alternating structure, with two substituents attached to each phosphorus atom. CTP is a non-conjugated organic-inorganic hybrid with good structural and thermal stability. CTP-based compounds have been widely used in applications involving high temperature resistance, UV light, nuclear radiation, flame retardancy, drug delivery, and fluorescence detection [30]. Hexachlorocyclotriphosphazene (HCCP) is the raw material most commonly used to synthesize CTP derivatives. Its P–Cl bond exhibits high reactivity and can easily be substituted by nucleophilic reagents (such as phenolic hydroxyl, amine, and thiol groups) under conditions involving organic or inorganic bases [31,32]. Additionally, CTP is composed of two kinds of flame-retardant elements, N and P. The synergistic flame-retardant effect of N and P contributes to superior flame retardancy compared to flame retardants containing only either N or P. Thus, CTP-based compounds are widely used as flame-retardant agents. However, there are relatively few reports on cyclotriphosphazene-containing thermosetting resins. According to the literature, the main phosphazene ring-containing thermosetting resins are epoxy and benzoxazine resins [33,34]. For phthalonitrile, the only report of a phosphazene ring-containing phthalonitrile (synthesized from HCCP, hydroquinone, and 4-nitrophthalonitrile) was by Zhao et al. in 2015. Their results showed that the curing reaction of CPCP could be triggered by adding 5 wt% 4-(hydroxylphenoxy)phthalonitrile, and the resulting cured CPCP (final curing procedure: 380 °C/2 h) exhibited a high *T*_g_ (>380 °C), a high decomposition temperature (the *T*_d,5%_ was 527 °C in N_2_), and a high char yield (83%) at 800 °C [35].

Herein, we further investigate the properties of cyclotriphosphazene-containing phthalonitrile resins and address the shortcoming of the slow curing reaction rate of phthalonitrile by designing and preparing a branched cyclotriphosphazene-containing phthalonitrile (CTP–PN) with self-catalysis. The curing reaction and processability of CTP–PN are studied along with the structures of CTP–PN and CTP–PN cured to varying degrees. The thermomechanical properties of cured CTP–PN (prepared using the hot press method) are also examined. In addition, the fractural morphology, thermal stability, and thermal decomposition of cured CTP–PN are analyzed and discussed.

## 2. Results and Discussion

### 2.1. Structure of CTP–PN

The structure of CTP–PN was characterized using FT–IR, ^1^H–NMR, 13 C–NMR, and Q–TOF–MS. The results are shown in Figure 1, and the structures of HCCP, 4-APN, and CTP–PN are presented in Figure 1. Figure 1a shows the FT–IR spectra of 4-APN, HCCP, and CTP–CN. Characteristic absorption peaks for 4-APN are observed at 2229 cm^−1^ and 3374–3453 cm^−1^, which correspond to nitrile and amine groups, respectively [36]. For HCCP, strong absorption bands corresponding to P=N, P–N, and P–Cl are observed at 1177 cm^−1^, 874 cm^−1^, and 594 cm^−1^, respectively [37,38]. After replacing the Cl atoms of HCCP with the amine groups of 4-APN, the absorption bands for –NH_2_ and P–Cl disappear. In addition, absorption bands for –CN and P=N can be observed in the products. Furthermore, the characteristic peak for P–N in HCCP at 874 cm^−1^ exhibits a blue shift to 949 cm^−1^ in the products, indicating the formation of the P–NH–Ph bond. Additionally, absorption bands of –NH– stretching vibration are observed at around 3337 cm^−1^ (a weak and wide absorption peak) and 1591 cm^−1^ [39,40]. Therefore, the FT–IR results indicate that the Cl atoms of HCCP were replaced with –NH_2_, forming a P–NH–Ph bond. Figure 1b shows the ^1^H–NMR spectrum (*δ*, ppm) of CTP–PN. Peaks at 2.50 ppm and 3.31 ppm are ascribed to the protons of the DMSO–d6 solvent and H_2_O [41]. Signals at 7.26–7.28 ppm, 7.39–7.41 ppm, 8.04 ppm, and 8.12–8.14 ppm correspond to the benzene protons [35,42,43]. In addition, the sharp peak at 7.83 ppm is attributed to the secondary amine protons of the P–NH–Ph bond. Referring to the ^13^C–NMR spectrum presented in Figure 1c, the chemical shifts at 107.67, 115.42, 115.45, 119.93, 116.68, 121.43, 122.06, 122.43, 136.32, 137.69, 152.67, and 161.34 ppm can be ascribed to carbon atoms in both benzene rings and nitrile groups [38,42,43]. In order to further characterize the structure of CTP–PN, mass analysis was performed using a Q–TOF–MS device. From the MS spectrum in Figure 1d, a protonated molecular ion peak at m/z = 236.0848 with 100% abundance can be observed. This is consistent with the molecular formula of C_14_H_8_N_3_O (m = 234.07). Therefore, the results of ^1^H–NMR, ^13^C–NMR, and MS analysis further support that CTP–PN was successfully synthesized from 4-APN and HCCP.

### 2.2. CTP–PN Curing Reaction and Processability 

The curing reaction and processability of CTP–PN were studied using DSC and DRA, and the results are shown in Figure 2. Figure 2a shows the DSC curves of CTP–PN. It can be seen that the DSC curves corresponding to different heating rates exhibit a wide and gentle endothermic peak at about 200 °C that does not obviously change as the rate of heating is increased. This endothermic peak can be ascribed to the melting process of CPT–PN [27]. In each DSC curve, an exothermic peak appears at 268–313 °C, which is attributed to the self-polymerization of CTP–PN. This occurs because the active hydrogen from the secondary amine proton of the P–NH–Ph bond catalyzes the curing reaction of the nitrile groups [44]. To further study the curing reaction and processability of CTP–PN, DRA was used to characterize the change in the viscosity of CTP–PN as a function of the temperature. It can be seen in Figure 2b that the CTP–PN viscosity slightly increases and then decreases between 70 °C and 90 °C, with similar changes occurring again between 90 °C and 135 °C. These changes are attributed to the softening of solid CTP–PN as the temperature increases. At temperatures ranging from 158 °C to 210 °C, the viscosity dramatically decreases from 12,000 Pa·s to 230 Pa·s, which corresponds to the melting process of CTP–PN. The viscosity then plateaus at about 210 °C before slowly decreasing to 100 Pa·s at 245 °C. Although the curing of CTP–PN occurs at 210 °C, the changes in viscosity at 210–245 °C can be attributed to the generation of only small amounts of crosslinking networks, which are insufficient to effectively restrict the mobility of molecular chains as the temperature increases. Above 245 °C, the curing reaction of nitrile groups is significantly promoted, leading to a rapid increase in viscosity [14,22]. Thus, the results of DSC and DRA demonstrate that CTP–PN possesses self-catalyzing properties and that the curing reaction of the nitrile group is initiated at a relatively low temperature (~210 °C).

### 2.3. Structure of Cured CTP–PN

To study the curing reaction of CTP–PN, FT–IR was used to characterize the structures of cured CTP–PN. The details of curing procedures are presented in Table 1. According to the DSC and DRA results, the curing reaction of CTP–PN is initiated at about 210 °C. Thus, 200 °C was selected as the initial curing temperature. As shown in Figure 3a, after curing at 200 °C for 2 h, characteristic absorption peaks for triazine rings (1359 cm^−1^) and phthalocyanine rings (3218 cm^−1^, 1078 cm^−1^, and 1011 cm^−1^) are generated. A weak absorption band of isoindoline at 1715 cm^−1^ is also detected. Thus, it can be concluded that the main structures generated from curing CTP–PN at 200 °C are triazine rings and phthalocyanine rings [21,44,45]. As the temperature is increased, the absorption intensity of the isoindoline and triazine rings exhibits a pronounced increase, while the absorption intensity corresponding to phthalocyanine rings decreases. This is because the higher temperature promotes the curing reaction of nitrile groups to form more isoindoline and triazine rings, so there is a decrease in the relative content of phthalocyanine rings in cured CTP–PN. Simultaneously, when the curing reaction of nitrile groups is occurring, the absorption peak corresponding to these groups is gradually reduced and has almost disappeared after curing at 280 °C. Therefore, following curing at higher temperatures, the structure of cured CTP–PN is predominantly composed of isoindoline and triazine rings. A characteristic absorption peak at 1200 cm^−1^ still exists in the cured CTP–PN, even after curing at 340 °C. This confirms that cyclophosphazene is not destroyed during curing. However, although the absorption intensity of the P–NH–Ph bond at 949 cm^−1^ decreases as the curing temperature increases, it does not disappear even under curing at 340 °C. This is because the secondary amino group partially participates in the curing reaction of nitrile groups [45], a phenomenon that was also observed by Yang’s group [14]. In summary, the FT–IR results confirm that the curing reaction of CTP–PN occurs at a relatively low temperature (200 °C). Moreover, below 260 °C, mainly phthalocyanine rings are formed from the curing reaction of nitrile groups. At higher temperatures (260–340 °C), this reaction is promoted, which results in the formation of isoindoline and triazine rings. The putative curing reaction of CTP–PN is presented in Figure 1.

### 2.4. Fracture Morphology of Cured CTP–PN

According to the FT–IR analysis of cured CTP–PN, the absorption intensity of nitrile groups obviously decreased after curing at 240 °C. This indicates that there is a significant occurrence of the curing reaction of CTP–PN. After further increasing the temperature to 280 °C, the absorption band corresponding to nitrile groups almost completely disappeared, implying a higher curing degree of CTP–PN. To study the influence of different curing degrees, SEM was used to scan the fracture morphology of cured CTP–PN. The fracture morphology of cured CTP–PN films (cured at 240 °C and 280 °C) is shown in Figure 4. Figure 4a,b show the fracture morphology for curing at 240 °C, where it can be seen that the cured CTP–PN film has a very smooth surface with no cracks or holes present. This is because, although the curing reaction of nitrile groups was efficiently triggered at 240 °C, the curing degree of nitrile groups remained relatively low, resulting in the formation of incomplete crosslinking networks. In Figure 4c, after curing at 280 °C, raised particles and shallow holes are observed on the surface. In Figure 4d, fracture stripes and a rough surface can be observed in the magnified photomicrographs. This indicates that the brittleness of CTP–PN can be alleviated to a certain extent by elevating the curing degree [15].

### 2.5. Thermal Stability of Cured CTP–PN

The thermal stability of cured CTP–PN (cured at 200 °C, 240 °C, and 340 °C, respectively) was studied based on TGA in a nitrogen atmosphere, and the results are presented in Figure 5a. The initial decomposition temperature of cured resins can be regarded as the temperature at which a weight loss of 5% (T_d5%_) occurs. It can be seen that the T_d5%_ of cured CTP–PN is above 360 °C. When the curing temperature is increased to 340 °C, the T_d5%_ is increased to 405 °C. Additionally, the char yield of cured CTP–PN exceeds 70% at 800 °C. Figure 5b shows the DTG curves and the temperature corresponding to the highest degradation rate (T_max_). Pure phthalonitrile resins usually exhibit outstanding thermal stability. For example, bisphenol A-based phthalonitrile cured at 280 °C has a T_d5%_ of 430 °C and a char yield of 61.4% at 800 °C [46]. Cured resorcinol-based phthalonitrile (cured at 375 °C for 5–8 h) has a T_d5%_ of 475 °C and a char yield of 72% at 800 °C [47,48]. However, after the incorporation of a phosphazene ring into phthalonitrile, there is a decrease in the thermal decomposition temperature. This is due to the decomposition of phosphazene rings at lower temperatures of 350–450 °C as a result of P–N bonds having lower thermal stability than C–C bonds [49]. With the increase in the curing temperature, the T_d5%_ of cured CTP–PN dramatically increases from 360 °C to 405 °C. This is ascribed to the higher curing degree of CTP–PN, which allows it to restrict the vibration of phosphazene rings and thus increase the decomposition temperature. At temperatures above 450 °C, there is decomposition of ether bonds and nitrogen-containing aromatic heterocycles. However, CTP–PN cured at lower temperatures exhibits a lower decomposition rate and higher char yield than CTP–PN cured at higher temperatures. This is because phosphazene decomposition occurs, which results in the generation of a phosphorous-rich char layer that reduces the thermal conductivity of the sample surface. Under these conditions, the char layer further retards the thermal decomposition of the underlying matrix by hindering heat transfer from the surface to the interior [50,51]. The data for the T_d5%,_ T_max,_ and char yield are shown in Table 2.

### 2.6. Thermomechanical Properties of Cured CTP–PN

The thermomechanical properties of cured CTP–PN (cured at 240 °C and 280 °C, respectively) were studied using TMA. As shown in Figure 6, CTP–PN cured at 240 °C exhibited an expansion ratio of 0.58% at 285 °C. Above this temperature, the expansion ratio did not obviously change until reaching 320 °C. With further temperature increases, the expansion ratio rapidly increased, indicating that there was a rapid expansion of cured CTP–PN. However, although the expansion ratio of cured CTP–PN slightly increased after curing at 280 °C, there was no obvious change as the temperature was increased to 350 °C. This confirms that as the curing temperature is increased, the higher curing degree could endow cured CTP–PN with more favorable thermomechanical properties. Additionally, the coefficient of linear thermal expansion (CTE) was calculated based on the TMA test results. The results showed that the CTE of CTP–PN cured at 240 °C is 25.9 ppm/°C at 270 °C and 24.7 ppm/°C at 285 °C. For CTP–PN cured at 280 °C, the CTE is 28.4 ppm/°C at 270 °C and 24.9 ppm/°C at 285 °C. Thus, the results of the TMA demonstrate that cured CTP–PN possesses excellent thermomechanical properties [4,15,52].

### 2.7. Thermal Decomposition Analysis of Cured CTP–PN

According to the TGA results, it can be seen that the initial decomposition of cured CTP–PN occurs at about 360 °C, and there is rapid decomposition at temperatures around 500 °C. Thus, in order to study the decomposition of CTP–PN, the decomposition products of cured CTP–PN (the maximum curing temperature was 280 °C) were obtained after heat treatments at 350 °C, 500 °C, and 800 °C for 5 min in a nitrogen atmosphere. The structures and elemental composition of decomposition products were studied using FT–IR and XPS, respectively. Figure 7 shows the FT–IR spectra of the decomposition products. When compared to cured CTP–PN, it can be seen that there are no obvious changes in the characteristic absorption bands for the phosphazene (949 cm^−1^ and 1200 cm^−1^), phthalocyanine (1011 cm^−1^ and 1078 cm^−1^), triazine (1359 cm^−1^), isoindoline (1715 cm^−1^), and benzene (1438 and 1061 cm^−1^) rings after treatment at 350 °C. This indicates that cured CTP–PN is stable below 350 °C. Then, after treatment at 500 °C, the absorption band at 949 cm^−1^ disappears, and there is an obvious chemical shift in the band at 1200 cm^−1^ corresponding to P=N, indicating that there is rapid decomposition of the phosphazene ring. However, although the bands for the isoindoline, triazine, and phthalocyanine rings have somewhat decreased absorption intensities, they can still be observed. This indicates that the nitrogen-containing aromatic heterocycles possesses excellent thermal stability. Moreover, it can be inferred that there is rapid decomposition of the benzene ring based on the decrease in the absorption intensity of the band at 1438 cm^−1^. Finally, following treatment at 800 °C, the characteristic absorption bands almost completely disappear due to the generation of carbon residue [53].

XPS was further applied to study the elemental composition of the decomposition products, and the results are shown in Figure 8. Figure 8a is the XPS survey spectrum of cured CTP–PN, which indicates the chemical composition of CTP–PN. As shown in Figure 8b, the C1s spectrum exhibits three peaks. The peak in the binding energy at 284.8 eV corresponds to the C=C bond of the benzene ring. The peak at 286.50 eV is assigned to a C–O bond, while the binding energy peak located at 288.30 eV corresponds to the N–C=N bond in the phthalocyanine, triazine, and isoindoline ring structures [54]. The N1s spectrum contains three peaks located at 398.47 eV (P–N=P), 400.09 eV (C=N–C), and 401.87 eV. The peaks assigned to P–N=P and C=N–C bonds are part of the cyclotriphosphazene structure and aromatic heterocycles containing nitrogen, respectively [55,56]. The binding energy peak at 401.87 eV is attributed to the hydrogen-bonded amine that forms between P–NH–Ph bonds [57]. As shown in Figure 8d, the P2p spectrum exhibits a single peak at 134.22 eV, indicating a P atom with the same chemical structure as in CTP–PN [55]. Hereby, the results of XPS analysis are consistent with the results of ^1^H–NMR and FT–IR presented above.

Figure 9 shows the high-resolution C1s, N1s, and P2p XPS spectra of the decomposition products of cured CTP–PN. It can be seen that after treatment at higher temperatures, the decomposition products are still composed of C, N, and P elements. As shown in Figure 9a, there is no change in the C1s, N1s and P2p spectra based on a comparison to cured CTP–PN. In particular, there is no obvious change in the ratio of each chemical bond. This means there is no decomposition of cured CTP–PN at 350 °C. However, when the temperature is increased to 500 °C, rapid decomposition occurs. It can be seen that there is a significant reduction in the intensity of peaks corresponding to the C–O–C and N–C=N bonds in the C1s spectrum in Figure 9b compared to the C1s spectrum in Figure 9a. At the same time, only a single peak at 399.75 eV remains in the high-resolution N1s spectrum. This indicates that there is complete decomposition of cyclotriphosphazene and P–NH–Ph bonds. Moreover, partial decomposition of nitrogen-containing aromatic heterocycles also occurs at 500 °C. Following the decomposition of cyclotriphosphazene and P–NH–Ph bonds, there is a slight shift in the peak of the P2p spectrum from 134.12 eV to 133.83 eV, which is due to the generation of phosphorous-containing char (P=N) and the decreased presence of the electron-withdrawing group (P–NH–Ph) attached to the P atom. In Figure 9c, the C1s, N1s, and P2p spectra of residual char at 800 °C indicate that it is mainly composed of P-containing graphitic C and N [54]. Therefore, based on the results from TGA together with the FT–IR and XPS analysis of the decomposition of cured CTP–PN, we propose a potential mechanism for thermal decomposition in nitrogen, which is presented in Figure 2 [51,58].

## 3. Materials and Methods

### 3.1. Materials

Hexachlorocyclotriphosphazene (HCCP, purity ≥ 99.0%) was purchased from Shanghai Haohong Biopharmaceutical Technology Co., Ltd., Shanghai, China. Potassium carbonate (K_2_CO_3_, AR, purity 99%) and acetonitrile (GC, purity > 99.5%) were supplied by Shanghai Aladdin Biochemical Technology Co., Ltd., Shanghai, China. Petroleum ether (60–90, AR) and ethyl acetate (AR) were purchased from Kelong reagent Co., Ltd., Chengdu, China. 4-Aminophenoxyphthalonitrile (4-APN) was synthesized in our lab according to a reported method [59]. All of the raw materials were used directly without purification.

### 3.2. Preparation of CTP–PN

Hexa(4-(3,4-dicyanophenoxy)phenylamino)-cyclotriphosphazene (CTP–PN) was synthesized from HCCP and 4-APN with K_2_CO_3_ as the acid-binding agent in an acetonitrile solvent. Acetonitrile (80 mL) was first added into a three-necked flask with magnetic stirring at room temperature. Then, 0.01 mol HCCP (3.48 g) and 0.06 mol 4-APN (14.1 g) were sequentially added into the acetonitrile solvent. After achieving the uniform mixing of HCCP and 4-APN, 0.035 mol K_2_CO_3_ (4.83 g) was added into the mixture solution. Then, the reaction temperature was heated to 50 °C (oil bath) for 8 h to allow for the completion of the reaction. Following this, the mixed CTP–PN solution was cooled to room temperature, poured into water with mechanical agitation, and then left overnight to allow for sedimentation. The block of solid CTP–PN that had settled at the bottom of beaker was then removed and crushed into powder. This CTP–PN powder was washed several times using deionized water, and dried CTP–PN powder was finally obtained after drying at 50–60 °C in a vacuum oven. The yield of CTP–PN following this step was 92.4%. In order to further remove unreacted HCCP and 4-APN, the dried CTP–PN powder was sequentially washed three times using petroleum ether and ethyl acetate. Purified CTP–PN was obtained after drying in a vacuum oven at 40–50 °C. The yield of CTP–PN following this step was 50.7%. The structure of CTP–PN and procedures for its preparation are presented in Figure 1.

### 3.3. Preparation of Cured CTP–PN

Cured CTP–PN with different curing degrees was prepared using a high-temperature oven. The curing procedures and corresponding sample numbers are presented in Table 1. A cured CTP–PN film was prepared using a hot press method according to the following preparation procedures. CTP–PN was pretreated at 200 °C for 20 min to obtain a CTP–PN prepolymer, which was then ground into powder and placed into a stainless steel mold (dimensions: 10 × 10 × 1 mm^3^ corresponding to width × length × thickness). The preparation pressure was 3–5 MPa, and curing procedures of 200 °C/2 h–220 °C/2 h–240 °C/2 h and 200 °C/2 h–220 °C/2 h–240 °C/2 h–260 °C/2 h–280 °C/2 h were used to prepare the cured CPT–PN films.

### 3.4. Characterizations

The structure of CTP–PN was characterized using Fourier transform infrared spectroscopy (FT–IR, iS50, Thermo, Waltham, MA, USA), ^1^H–NMR, and ^13^C–NMR (Bruker NMR spectrometer, 400 MHz, Billerica, MA, USA). CD_6_SO (DMSO–d6) was chosen as the solvent for NMR tests. The mass analysis of CTP–PN was conducted using high-resolution mass spectrometry (HRMS, Aglient 6546 Q–TOF MS, Boulder, CO, USA, positive ion mode), with ESI chosen as the ion source. Differential scanning calorimetry (DSC, DSC 200 F3, Netzsch, Germany) with heating rates of 5, 10, 15, and 20 °C/min was applied to characterize the curing reaction of CTP–PN from 50 °C to 350 °C. The processability of CTP–PN was studied using dynamic rheological analysis (DRA, Discovery HR–20, TA Instrument, Phoenix, AZ, USA). Complex viscosity changes were recorded in the temperature range of 70–300 °C (5 °C/min) in an air atmosphere. The fractural morphology of the cured CPT–PN film was investigated using scanning electron microscopy (SEM, Phenom pharos, Eindhoven, Netherlands) with an operation at 15 kV. The thermal stability of cured CTP–PN was characterized using thermogravimetry (TGA, TG 290 F3, Netzsch, Germany) from 50 °C to 800 °C with a heating rate of 20 °C/min in a nitrogen atmosphere. The elemental composition and residues of cured CTP–PN (obtained through heat treatment in a nitrogen atmosphere) were characterized using X-ray photoelectron spectroscopy (XPS, K–Alpha, Thermo Fisher Scientific, Waltham, MA, USA), with C1s = 284.8 eV taken as the calibration binding energy. The thermal–mechanical properties of the cured CTP–PN film were characterized using thermal mechanical analysis (TMA, TMA–60, Shimadzu, Japan) from 50 °C to 350 °C with a heating rate of 5 °C/min. The dimensions of the sample for TMA characterization were about 3 × 3 × *d* (width × length × thickness, mm).

## 4. Conclusions

A branched phthalonitrile-containing cyclotriphosphazene (CTP–PN) was synthesized from HCCP and 4-APN. The structure of CTP–PN was confirmed based on FT–IR and ^1^H–NMR, ^13^C–NMR, and MS analyses. The results of the XPS analysis of cured CTP–PN revealed the existence of the P–NH–Ph bond in CTP–PN and further confirmed its successful synthesis. The presence of P–NH–Ph endows CTP–PN with the property of self-catalysis, and we found that the curing reaction of nitrile groups is initiated at 200 °C. CTP–PN exhibits a lower softening temperature (100 °C), and the lowest viscosity (100 Pa·s) could be obtained at 245 °C. At temperatures higher than 245 °C, there is a dramatic increase in viscosity due to the acceleration of the curing reaction. In cured CTP–PN, mainly structures composed of triazine rings and phthalocyanine rings are generated at low curing temperatures (200 °C–240 °C). After curing at higher temperatures, mainly structures composed of isoindoline and triazine rings are generated. Moreover, cured CTP–PN exhibits favorable thermomechanical properties (CTE: 25 ppm/°C at 285 °C). A *T*_d5%_ of up to 405 °C can be obtained for cured CTP–PN, and its char yield at 800 °C exceeds 70% in N_2_. The results of thermal decomposition analysis revealed that cured CTP–PN is stable below 350 °C. Above this, there is decomposition first of cyclotriphosphazene and P–NH–Ph and C–O–C bonds (350–500 °C) and then nitrogen-containing aromatic heterocycles (>500 °C), ultimately resulting in the formation of graphitic residual char with P and N elements. Thus, CTP–PN has the potential to be used as a high-temperature-resistant material with a low CTE.

## Data Availability

Data will be made available on request.

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
