# Peer review of "Synthesis and Investigation of the Properties of a Branched Phthalonitrile Containing Cyclotriphosphazene"

_molecules, 2024, doi:10.3390/molecules29235668_

Round 1
Reviewer 1 Report
Comments and Suggestions for Authors
a) A thorough grammar check is required for the whole manuscript.
b) Experimental section -please explain with more clarity about the curing procedure Line 376, 377
c) Please label the major peaks in fig 1a with the respective wave numbers (that makes it easier for readers).
d) Need references for FTIR peaks as well as 1H-NMR
e) Need a reference for line 165-167
f) How did you obtain the fracture surface for SEM morphology? Because the fracture surface is relatively smoother (Figures 4a and 4b) than the other one (Figures 4c and 4d), you cannot conclude that the one is brittle. It requires some mechanical testing (stress -strain graphs).
g) Line 225-227 “Additionally, no phase separation could be observed from the SEM photomicrographs. This indicated that CTP exhibited good compatibility with phthalonitrile.”
SEM is mostly used for topography and surface features; it is hard to arrive at a conclusion on phase separation or phase difference on molecular level. Ideally, a TEM image can be obtained to determine if there is any phase difference or other imaging techniques such as AFM phase or resonance modulation imaging, etc
h) It would be ideal to have a derivative weight curve (dm/dTmax) for the TGA. What was the maximum degradation temperature for various samples? A table showing Td5% and Tmax (the temperature at which the degradation rate is fastest) would be useful for the manuscript.
i) In conclusion, you have mentioned that the current materials have the potential as flame retardant. Have you done any flammability tests in addition to the thermal stability test? Unless it is tested, you cannot conclude in this way
Author Response
- a) A thorough grammar check is required for the whole manuscript.
Responses: Thanks for your works and comments on manuscript. According to your comments, we have carefully checked the grammar in the whole manuscript and corrected the grammar mistakes. To facilitate reviewing, all of the modifications and changes in revised manuscript were marked as Red.
- b) Experimental section -please explain with more clarity about the curing procedure Line 376, 377.
Responses: We are sorry for making you feel confusion here. The curing procedure in line 376 and 377 represented that after the CTP-PN prepolymer powder was put into stainless steel mold, the stainless steel mold was placed into parallel-plate hot-press machine at 200 oC. After that, the pressure imposed on stainless steel mold was gradually increased to 3-5 MPa. Then, the cured CTP-PN films were obtained with the temperature procedures of “200 oC/2 h-220 oC/2 h-240 oC/2 h” and “200 oC/2 h-220 oC/2 h-240 oC/2 h-260 oC/2 h-280 oC/2 h”, respectively.
- c) Please label the major peaks in fig 1a with the respective wave numbers (that makes it easier for readers).
Responses: According to your suggestion, the major peaks in Figure 1(a) were labelled with wave numbers in revised manuscript and copied as follows.
Figure 1. (a) FT-IR spectrum of 4-APN, HCCP and CTP-PN, (b) 1H-NMR spectrum of CTP-PN, (c) 13C-NMR spectrum of CTP-PN and (d) TOF-MS spectrum of CTP-PN.
- d) Need references for FTIR peaks as well as 1H-NMR.
Responses: According to your comments, references [36], [37] and [38] (line 128-130 in revised manuscript) for FTIR peaks and references [35], [41], [42] and [43] (line 139-140 in revised manuscript) for 1H-NMR were added in revised manuscript, respectively.
- e) Need a reference for line 165-167.
Responses: References [14] and [22] was supplemented at the end of sentence “…was intensively promoted, leading to the rapid increase of viscosity [14, 22]” (line 178 in revised manuscript).
- f) How did you obtain the fracture surface for SEM morphology? Because the fracture surface is relatively smoother (Figures 4a and 4b) than the other one (Figures 4c and 4d), you cannot conclude that the one is brittle. It requires some mechanical testing (stress -strain graphs).
Responses: Thanks for your meaningful comments on the explainations about the brittleness of cured resin (judged by SEM). The fracture surface used to SEM observation was obtained by breaking the cured CTP-PN film at room temperature. From the SEM morphology, it can be seen that fracture surface of CTP-PN cured at 280 oC (Figure 4a and Figure 4b) was rougher than the CTP-PN cured at 240 oC (Figure 4c and Figure 4d). It can be concluded that the brittleness of cured CTP-PN could be partly improved as the temperature raised. But, it cannot conclude that cured CTP-PN possessed high brittleness, which was only judged by the fracture surface. Thus, according to your comments, the descriptions on the brittleness of cured CTP-PN were modified in revised manuscript and copied as follows. The mechanical properties will be further investigated in our following researches.
According to the FT-IR analysis on cured CTP-PN, after cured at 240 oC, absorption intensity of nitrile groups was obviously decreased and generated structures were main phthalocyanine ring. This indicated that curing reaction of CTP-PN was dramatically occurred. Then, with the further increasing of temperature to 280 oC, absorption band of nitrile group almost disappeared, implying the higher curing degree of CTP-PN. Therefore, in order to study the influence of different curing degree on cured CTP-PN, SEM was used to scan the fracture morphology of cured CTP-PN. Fracture morphology of cured CTP-PN films (cured at 240 oC and 280 oC, respectively) was shown in Figure 4. Figure 4(a) and Figure 4(b) were the fracture morphology of CTP-PN film cured at 240 oC. It can be seen that cured CTP-PN exhibited very smooth surface and no cracks or holes existed on surface. This was due the reason that although curing reaction of nitrile groups was efficiently triggered at 240 oC, curing degree of nitrile groups was relatively low and formed incomplete crosslinking networks. In Figure 4(c), after cured at 280 oC, raised particles and shallow holes could be observed on the surface of cured CTP-PN. In Figure 4(d), fracture stripes and roughed surface could be observed in the magnified photomicrographs. This indicated that brittleness of CTP-PN could be improved to a certain extent by elevating the curing degree.
- g) Line 225-227 “Additionally, no phase separation could be observed from the SEM photomicrographs. This indicated that CTP exhibited good compatibility with phthalonitrile.” SEM is mostly used for topography and surface features; it is hard to arrive at a conclusion on phase separation or phase difference on molecular level. Ideally, a TEM image can be obtained to determine if there is any phase difference or other imaging techniques such as AFM phase or resonance modulation imaging, etc
Responses: Your comments are absolutely correct and we quite agree with you. Based on your comments, the descriptions on phase separation analysis were removed in revised manuscript. The other methods on studying phase separation you suggested will provided us new ways to characterize the phase separation of cured matrix resins.
- h) It would be ideal to have a derivative weight curve (dm/dTmax) for the TGA. What was the maximum degradation temperature for various samples? A table showing Td5% and Tmax (the temperature at which the degradation rate is fastest) would be useful for the manuscript.
Responses: According to your suggestions, derivative weight curve (DTG, Figure 5b in revised manuscript) for the TGA curves as well as a table showing Td5% and Tmax (Table 1 in revised manuscript) were supplemented in revised manuscript. DTG curves, data of Td5%, Tmax and char yield were copied as follows.
Figure 5. (a) TGA curves of cured CTP-PN and (b) DTG curves.
Table 1. Curing procedures of CTP-PN.
Cured CTP-PN |
Td5%/oC |
Tmax1/oC |
Tmax2/oC |
char yield at 800 oC |
Cured at 200 oC |
360 |
384 |
509 |
75.6% |
Cured at 240 oC |
376 |
380 |
509 |
72.8% |
Cured at 340 oC |
405 |
- |
509 |
70.0% |
- i) In conclusion, you have mentioned that the current materials have the potential as flame retardant. Have you done any flammability tests in addition to the thermal stability test? Unless it is tested, you cannot conclude in this way.
Responses: Thanks for your useful suggestions. We have deleted the description of CTP-PN has the potential as flame retardant in Conclusion section in revised manuscript.

Reviewer 2 Report
Comments and Suggestions for Authors
The researchers detailed the synthesis of a branched phthalonitrile with cyclotriphosphazene (CTP-PN) using HCCP and 4-APN. They provided a comprehensive characterization of the material. However, several issues need to be addressed before the work can be considered for publication:
1. While the authors presented phthalonitrile-containing cyclotriphosphazene as a potential high-temperature resistant material with low CTE and flame-retardant properties, the claim in line 100 referring to "flame-retardant elements Phosphorous" is questionable. Phosphorus in its elemental form is highly flammable, so this statement requires justification. (Although phosphorus-based flame retardants exist, elemental phosphorus is highly combustible).
2. Much of the introduction requires rephrasing, particularly lines 31 to 35.
3. For discussions on Cyclotriphosphazene, it is recommended to cite more recent references, such as: J. J. C. Lee, M. H. Chua, S. Wang, Z. Qu, Q. Zhu, J. Xu, Chem. Asian J. 2024, 19, e202400357.
4. 1. Given that researchers have anticipated the potential release of HCN during thermal breakdown, the application of this substance presents a significant safety hazard.
5. Reference 50 needs correction (it is incomplete).
Comments on the Quality of English LanguageThe quality of the English needs to be proved.
Author Response
- While the authors presented phthalonitrile-containing cyclotriphosphazene as a potential high-temperature resistant material with low CTE and flame-retardant properties, the claim in line 100 referring to "flame-retardant elements Phosphorous" is questionable. Phosphorus in its elemental form is highly flammable, so this statement requires justification. (Although phosphorus-based flame retardants exist, elemental phosphorus is highly combustible).
Responses: Thanks for your works and comments on our manuscript and we appreciate your rigorous attitude towards writing research paper. According to your comments, the description “The synergistic flame-retardant effect of N and P contributed to better flame retardancy than single N or P” has modified as “The synergistic flame-retardant effect of N and P contributed to better flame retardancy than single N or P-containing flame retardants” (line 102-104 in revised manuscript).
- Much of the introduction requires rephrasing, particularly lines 31 to 35.
Responses: Thanks for your suggestions on the writing of manuscript. The writing of introduction in line 31-35 (line 25-38 in revised manuscript) as well as the other descriptions in revised manuscript were checked and modified.
- For discussions on Cyclotriphosphazene, it is recommended to cite more recent references, such as: J. J. C. Lee, M. H. Chua, S. Wang, Z. Qu, Q. Zhu, J. Xu, Chem. Asian J. 2024, 19, e202400357.
Responses: This literature reviewed the synthesis of CP-based small molecules through nucleophilic substitution of HCCP, and their applications, including flame retardants, liquid crystals (LC), chemosensors, electronics, biomedical materials, and lubricants. Thus, it is essential for us to know the development of cyclotriphosphazene based materials. This literature was cited in Introduction section and the number of this reference is [30] in revised manuscript (line 98 in revised manuscript).
- Given that researchers have anticipated the potential release of HCN during thermal breakdown, the application of this substance presents a significant safety hazard.
Responses: We agree with your opinion on the application of this material. Before the application of this substance, the decomposition of this substance must be carefully investigated to assess the influence on human health.
- Reference 50 needs correction (it is incomplete).
Responses: Reference 50 (Reference 58 in revised manuscript) has been modified (line 562-563 in revised manuscript).
Reviewer 3 Report
Comments and Suggestions for Authors
The authors synthesised a branched, potentially high-performance polymeric material, hexa(4-(3,4-dicyanophenoxy)phenylamino)-cyclotriphosphazene (CTP-PN), starting from commercially available substrate and 4-aminophenoxyphthalonitrile, synthesised by them applying known methods. The structure of the product was confirmed by basic spectroscopic methods (FT-IR and 1H-NMR). X-ray photoelectron spectroscopy (XPS) applied to the product supported its structural analysis and enabled the primary characterisation of its thermal behaviour under the curing process. The authors also present more advanced physicochemical studies on the title material, applying DSC, TG, SEM, thermal mechanical (TMA) and dynamic rheology (DRA) techniques.
In this work, the authors undertook the task of improving the molecular systems known in the literature. They employed various physicochemical methods, which were adequately selected for this purpose. However, the methodological description is generally not very precise, and the analysis of the results provided by the authors left the impression that they sometimes overinterpret the presented data. The reliable support of the literature on the subject helps this work. The introduction is comprehensive; it reviews essential works for the investigated issues, including many newer ones.
Below are the flaws in the submitted work, that should be improved/corrected:
1. Abstract. So many abbreviations regarding techniques specific to a given field cannot be used. This is a part to be read by various scientists, including those not specialised in the area. Following that, the full names should be used in this part. The justification for choosing the subject of the investigations should be briefly presented here.
2. For the paper's clarity, it is beneficial to present the structures of studied compounds at the beginning of the Results and Discussion. Such a presentation with clear and correct numbering of atoms will enhance the paper's clarity.
3. The presented 1H NMR spectra are not very informative. The scale is too broad, and significant residual signals arise from solvent and water. The authors can improve the reliability of the assignments by presenting extended diagnostic ranges (e.g. aromatic region) and signal integrations (e.g. in ESI). Recording and resolving 1H-1H COSY, 13C NMR, and 1H-13C HETCOR spectra will further enhance the quality of the work.
4. It is of utmost importance to provide a mass analysis (HQ MS) of the product of synthesis (presented in Fig. 1). This is crucial for unambiguously identifying the quite complex molecular system under study. At the very least, the parent ion (recorded by, e.g. ESI-QTOF technique) should be recorded to support spectroscopic analyses, which alone are not sufficient for unambiguous identification of the product.
5. In my opinion, the spectroscopic experiments do not indicate the advanced conclusions the authors propose in the Results concerning the curing reaction of CTP-PN (Scheme 1, p. 6). Other works seem to support the provided interpretations, but they are presented very briefly. Please, comment on this.
6. XPS experiments must be described precisely in terms of technique - more information should be provided in the Materials and Methods section. They comprise much of the research work and strongly contribute to the conclusions. Were XPS experiments repeated or pre-calibrated? Obtaining repeatability based on such data would be challenging if a single measurement was performed for each case. It would be beneficial to present additional measurements for data verification.
Comments on the Quality of English LanguageLanguage and editing errors, style, and grammar must be improved by a native speaker or by applying a professional linguistic tool.
Author Response
- Abstract. So many abbreviations regarding techniques specific to a given field cannot be used. This is a part to be read by various scientists, including those not specialised in the area. Following that, the full names should be used in this part. The justification for choosing the subject of the investigations should be briefly presented here.
Responses: Thanks for your comments and useful suggestions on our manuscript. According to your suggestions, all of the abbreviations mentioned in Abstract section was supplemented with full names. For this work, after reviewed the literatures of phthalonitrile resins, we found that preparation and investigation on the properties of cyclotriphosphazene (CTP)-containing phthalonitrile was relatively few. Thus, we designed and synthesized a kind of branched phthalonitrile containing CTP. This was the justification for this work. The justification for choosing the subject of the investigations was briefly presented in the first sentence of Abstract section and copied as follows. To facilitate reviewing, all of the modifications and changes in revised manuscript were marked as Red.
In order to study the properties of cyclotriphosphazene (CTP)-containing phthalonitrile, a kind of branched phthalonitrile containing CTP (CTP-PN) with self-catalysis was designed and synthesized.
- 2. For the paper's clarity, it is beneficial to present the structures of studied compounds at the beginning of the Results and Discussion. Such a presentation with clear and correct numbering of atoms will enhance the paper's clarity.
Responses: Your suggestion was very useful. Structures of hexachlorocyclotriphosphazene (HCCP), 4-aminophenoxyphthalonitrile (4-APN) and branched phthalonitrile containing CTP (CTP-PN) were presented in Scheme 1, which was located at the end of first paragraph of Results and discussions section.
- The presented 1H NMR spectra are not very informative. The scale is too broad, and significant residual signals arise from solvent and water. The authors can improve the reliability of the assignments by presenting extended diagnostic ranges (e.g. aromatic region) and signal integrations (e.g. in ESI). Recording and resolving 1H-1H COSY, 13C NMR, and 1H-13C HETCOR spectra will further enhance the quality of the work.
Responses: According to your comments, it can be known that you are the organic synthesis expert and good at characterizing the chemical structures of organic compounds. Your comments provide us more methods to study and confirm the chemical structures as designed. Thus, we quite appreciate your comments and suggestions on the chemical structure characterization. Based on your comments, we have narrowed the scale of chemical shift for 1H-NMR spectrum to highlight the aromatic region. Modified 1H-NMR spectrum graph was presented in Figure 1(b) in revised manuscript. Besides, 13C-NMR spectrum of CTP-PN was recorded and presented in Figure 1(c) in revised manuscript. From the 13C-NMR spectrum, it can be seen that the chemical shifts at 107.67, 115.42, 115.45, 119.93, 116.68, 121.43, 122.06, 122.43, 136.32, 137.69, 152.67 and 161.34 ppm were ascribed to carbon atom on benzene ring and nitrile groups, respectively. Thus, 13C-NMR spectrum provided another evidence to confirm the structure of CTP-PN. Modified 1H-NMR spectrum as well as 13C-NMR spectrum were copied as follows.
Figure 1. (b) 1H-NMR spectrum and (c) 13C-NMR spectrum.
- 4. It is of utmost importance to provide a mass analysis (HQ MS) of the product of synthesis (presented in Fig. 1). This is crucial for unambiguously identifying the quite complex molecular system under study. At the very least, the parent ion (recorded by, e.g. ESI-QTOF technique) should be recorded to support spectroscopic analyses, which alone are not sufficient for unambiguous identification of the product.
Responses: The characterization methods your provided greatly helped us to enhance the quality of the work. Based on comments, a mass analysis of CTP-PN was characterized by High Resolution Mass Spectrometry (HRMS, Aglient 6546 Q-TOF MS, positive ion mode) and ESI was chosen as ion source. Results of MS spectrum were presented in Figure 1(d) in revised manuscript and copied as follows. From the mass spectrum, it can be seen that protonated molecular ion peak at m/z=236.0848 with abundance of 100% could be observed. This was well consistent with the molecular formula of C14H8N3O (m=234.07). Therefore, MS analysis provided anther evidence to support spectroscopic analyses.
Figure 2. MS spectrum of CTP-PN.
- 5. In my opinion, the spectroscopic experiments do not indicate the advanced conclusions the authors propose in the Results concerning the curing reaction of CTP-PN (Scheme 1, p. 6). Other works seem to support the provided interpretations, but they are presented very briefly. Please, comment on this.
Responses: We are sorry for making you feel confusion here. We would like to make explanations on the curing reaction of nitrile group, which was catalyzed by active hydrogen from secondary amino group. Yang et al. investigated the curing reaction of aromatic nitrile-based resins containing both phthalonitrile and dicyanoimidazole groups (PNDCI) and used FT-IR to characterize the generated structures. Results showed that the characteristic absorption peak of N-H bond was disappeared after the curing reaction. They pointed out that the N-H may be involved in the curing reaction of nitrile groups. For the cured PNDCI, the generated structures were mainly triazine
and isoindoline These results were different from the previous studies reported that phthalocyanine, triazine and isoindoline were the major components of phthalonitrile resins [1]. Wang et al. synthesized two kinds of secondary amino group-containing phthalonitrile resins, they used FT-IR to study the structures of cured phthalonitrile resins. They found that the generated structures were isoindoline, triazine ring and phthalocyanine ring. Besides, they also observed the decrease of absorption intensity of N–H bond [2,3]. Thus, in this work, FT-IR could be utilized to characterize the generated structures and analysis the changes of generated structures changed with different curing degree. Besides, relevant literatures were added in 2.3 section in revised manuscript.
References:
[1] 14. W. Yang, J. Qi, W. Tan, Z. Zhu, X. He, K. Zeng, J. Hu, G. Yang, Study on aromatic nitrile-based resins containing both phthalonitrile and dicyanoimidazole groups, Polymer. 2022, 255.
[2] 44. T. Wang, Z.-l. Wang, A.Q. Dayo, C.-y. Shi, H.-b. Liu, Z.-c. Pan, A.A.K. Gorar, J. Wang, H. Zhou, W.-b. Liu, Synthesis and properties of a novel autocatalytic phthalonitrile monomer and its copolymerization with multi-functional fluorene-based benzoxazine monomers, J. Appl. Polym. Sci. 2022, 139(21).
[3] 45. T. Wang, A.Q. Dayo, Z.-l. Wang, H.-m. Lu, C.-y. Shi, Z.-c. Pan, J. Wang, H. Zhou, W.-b. Liu, Novel self-promoted phthalonitrile monomer with siloxane segments: synthesis, curing kinetics, and thermal properties, New J. Chem. 2022, 46(9), 4072-4081.
- 6. XPS experiments must be described precisely in terms of technique - more information should be provided in the Materials and Methods section. They comprise much of the research work and strongly contribute to the conclusions. Were XPS experiments repeated or pre-calibrated? Obtaining repeatability based on such data would be challenging if a single measurement was performed for each case. It would be beneficial to present additional measurements for data verification.
Responses: Thanks for your comments. According to your comments, the detailed conditions for XPS tests were presented as follows. XPS experiments were performed on K-ALPHA device, which is provided by Thermo Fisher. The excitation source is Al ka and the energy of photons (hv) is 1486.8 eV with 400 μm-width beam. The vacuum pressure of analysis room is 2×10-9 mbar. Operating voltage and electric current were 15 kV and 10 mA. The energy of survey spectrum scanning and high-resolution elemental spectra were 150 eV (scanning step: 1 eV) and 50 eV (scanning step: 0.1 eV, number of cycling scanning times > 5). C1s=284.8 eV is taken as the calibration binding energy. Based on your comments, XPS was used to characterize the chemical composition of cured CTP-PN and cured CTP-PN matrix after treated at 350 oC once again. Results were presented as follows. It can be seen that the binding energy of C, N and P was well consistent with the results in manuscript. Herein, utilization of XPS to characterize the elemental composition could be reliable in this work.
Figure 3. XPS survey spectrum of cured CTP-PN and high-resolution elemental spectra of cured CTP-PN: (b) C1s, (c) N1s and (d) P2p.
Figure 4. XPS survey spectrum of cured CTP-PN after treated at 350 oC and high-resolution elemental spectra of cured CTP-PN: (b) C1s, (c) N1s and (d) P2p.

Round 2
Reviewer 3 Report
Comments and Suggestions for Authors
The authors have considered the reviewer's comments to a satisfactory degree. Where indicated, new measurements have been performed and commented on. Several new literature references have been added, which are essential in the presented context.
The current work in its present form is, in my opinion, suitable for publication in the Molecules journal.
Author Response
Comments and Suggestions for Authors
The authors have considered the reviewer's comments to a satisfactory degree. Where indicated, new measurements have been performed and commented on. Several new literature references have been added, which are essential in the presented context.
The current work in its present form is, in my opinion, suitable for publication in the Molecules journal.
Responses: Thanks for your acceptance on our revised manuscript. Thank you once again for your works on our manuscript.